# Determination of Water Content in Direct Resin Composites Using Coulometric Karl Fischer Titration

**DOI:** 10.3390/ma15238524

**Published:** 2022-11-30

**Authors:** André Faria-e-Silva, Lea Heckel, Renan Belli, Ulrich Lohbauer

**Affiliations:** 1Department of Dentistry, Federal University of Sergipe, Aracaju 49100-000, Brazil; 2Research Laboratory for Dental Biomaterials, Dental Clinic 1—Operative Dentistry and Periodontology, Friedrich-Alexander University in Erlangen-Nürnberg, 91054 Erlangen, Germany

**Keywords:** Karl Fischer titration, resin composite, water sorption

## Abstract

This study evaluated the water content and sorption of direct composites over 60 days using coulometric Karl Fischer titration (KFT). Plate-shaped specimens (10 × 10 × 1 mm^3^ of thickness) were built up using the composites Clearfil Majesty Posterior (CM), Grandio SO (GS), and Filtek Supreme XT (FS). Water contents were determined in non-stored specimens (control) or after storage in distilled water for up to 60 days (*n* = 5). The amount of water transferred from the specimens heated at 200 °C (isothermal mode) was measured in the Coulometer. The water content of non-stored specimens ranged from 0.28 to 1.69 wt% (5.6 to 31.2 μg/mm^3^) for GS and FS, respectively. The highest values of water sorption were observed for FS (25.3 μg/mm^3^ after 60 days). GS and CM showed similar water sorption after 60 days (≈9 μg/mm^3^), but an ultimate higher water content was observed for CM (0.9 wt%; 22.0 μg/mm^3^) than GS (0.7 wt%; 14.8 μg/mm^3^). Except for CM, no significant water sorption was observed between 21 and 60 days of storage. Since all composites presented some base water content, water sorption data alone do not account for the ultimate water content in direct resin-based composites.

## 1. Introduction

Direct restorations with resin composite are largely used in dentistry due to their reduced cost compared to indirect approaches and adequate esthetical and mechanical properties to restore anterior and posterior teeth. Clinical trials have reported annual failure rates of direct restorations using composites of approximately 2.0% [1]. However, these rates might exceed 5.0% when more pragmatic scenarios are evaluated, such as including high-risk patients [2]. Fractures of restoration or tooth structure and secondary caries are the main reasons for replacing a composite restoration in posterior teeth [3]. However, it is important to emphasize that the diagnosis of secondary caries is a challenge. Re-interventions in restorations done presently on stained margins are commonly done based on a misdiagnosis of secondary caries or microleakage [1,4,5].

Regarding restorations in anterior teeth, esthetic demands, including discoloration of restorations, are the leading cause of re-interventions [6]. Both stained margins and color changes in composite restorations are strongly related to the ability of resin-based materials to absorb water and staining solutions such as coffee, red wine, and others [7]. Moreover, water sorption can compromise the mechanical performance of the composite and favor the occurrence of fractures in the restoration [8,9].

Studies have evaluated the ability of dental composites to absorb water using the ISO 4049 standard [10,11,12,13]. This method estimates the water sorption based on the mass gain of the specimens due to the water storage [14]. Likely, mass loss on drying is used to calculate the water content. However, it is impossible to assure that the mass changes were caused by the gain or loss of water since composites present other leachable components.

Water molecules in polymeric materials can be classified as free and bounded. In the former state, water molecules occupy free volumes or nano-pores and can be completely released at the same temperature at which they are absorbed [15]. In the latter case, hydrogen bonds need to be broken down to release the bounded water, and higher temperatures are required to provide the energy demanded by this process [15]. The initial weight of specimens is critical when water sorption is measured using the ISO 4049 standard. Thus, weight is only determined after keeping them in a desiccator at 37 °C for 22 h, followed by a second desiccator (23 °C) for a further two hours [14]. These relatively low temperatures are likely unable to remove bounded water from bulk specimens. Therefore, misinterpretations might occur for materials with similar water sorption values but different base water content, which is unknown when ISO 4049 standard is used. On the other hand, Karl Fischer titration (KFT) allows measuring the evaporated water from heated specimens, increasing the method’s reliability since both free and bounded water are considered [16]. In the vaporization method, the pre-weighted specimens are heated inside a sealed vial, and water vapor is transferred to the titration cell through an inert gas or dried air stream [16]. Therefore, the present study aimed to measure the initial water content and amount of water absorbed by direct resin composites using coulometric KFT.

## 2. Materials and Methods

The dependent variable assessed was water content according to independent variables ‘composite’ (three levels) and ‘storage time’ (six levels). The manufacturers, monomeric and filler composition, and densities of composites studied are presented in Table 1. Water content was measured at 0 h (base measurement) and after 1, 2, 7, 21, or 60 days in distilled water (*n* = 5).

Plate-shaped specimens were built-up by inserting a single composite increment into a silicone matrix (10 × 10-mm, 1.3-mm-thick) between polyester strips. Composite was light-cured using the LED-based light-curing unit Bluephase G4 (Ivoclar Vivadent, Schaan, Liechtenstein). This unit had an emission spectrum with two-wavelength peaks (408 and 452 nm) and irradiance of 1200 mW/cm^3^ using the mode “high.” The active tip diameter of the light-curing unit is 8.8 mm, and it was positioned approximately 2 mm distant from the specimen to allow the light to reach its entire surface. Two 20 s photoactivations were performed on one side of the specimens. A further 20 s photoactivation was carried out on the opposite side of the specimen, totalizing 60 s of the light-curing procedure. 

The polymerized specimens were grounded with a fine diamond wheel (D50) to obtain square cross-sections of 10 × 10 × 1-mm^3^ and polished with SiC sandpaper (granulation #2000). Debris was removed with an ultrasound bath for 15 min, and the specimens were stored in a dry environment at room temperature for at least one week before the experiments. Water sorption of composites was assessed by storing the specimens in distilled water for 1, 2, 7, 21, or 60 days. We placed the specimens into plastic containers with bottoms covered with a sponge filled with distilled water. The sponge served as a carrier to improve the water contact with the entire specimen, which was stored in an oven at 37 ± 1 °C. Before the water content assessment, the excess water covering the specimens was removed using wipes (Kimtech Science, Ref. # 13292, Reigate, UK). Then, the specimens were weighed into 6 mL sample vials (Metrohm, Ref. # 6.2419.007, Herisau, Switzerland), which were hermetically sealed using aluminum septum caps.

The water content of specimens was determined using an oven sample processor (860 KF Thermoprep, Metrohm, Herisau, Switzerland) connected with a KF coulometer (917 Coulometer, Metrohm, Tokyo, Japan). The residual water of the system was removed (system conditioning), and the blank value was determined using an empty sealed vial. A solid standard for the KF oven method (Water standard oven 1%, Aquastar^®^, lot # FN1455454, Merck KGaA, Darmstadt, Germany) was used to check the accuracy of the specimens. The sealed vials containing the weighted specimens were placed in the oven. The specimens were heated isothermally at 200 °C, and water evaporated from them was transferred to the titration cell using a dried gas flow under a constant flow of 80 mL/min.

Titrations were carried out using a diaphragm-free cell to generate iodine at a constant current of 10 µA. The endpoint was set at a voltage of 50 mV and a stop drift of 5 µg/min, which is the amount of KF reagent per unit of time consumed. The coulometer automatically calculated the percentage of water content based on the specimen’s weight. The weight of water content per specimen’s volume was calculated and recorded in μg/mm^3^. Values in μg/mm^3^ were calculated to compare with those observed in prior studies using the ISO 4049 standard. Figure 1 illustrates the experimental setup, which followed the parameters reported in a prior study [17].

Water content data were analyzed by 2-way Analysis of Variance (ANOVA). Pair-wise comparisons were performed using Tukey’s test. The significance level was set at 95% for all analyses.

## 3. Results

Either measuring the water content by wt% or μg/mm^3^, 2-way ANOVA showed that the independent variables ‘composite’ (*p* < 0.001) and ‘storage time’ (*p* < 0.001) affected the water content. The interaction between the variables was also significant (*p* < 0.001). The results are presented in Table 2 and Table 3, and Figure 2. All materials presented some water content before the water storage, ranging from 0.28 to 1.69 wt% (5.6 to 31.2 μg/mm^3^) for Grandio SO and Filtek Supreme XT, respectively. Regardless of the storage time, the highest values of water content (1.69 to 3.07 wt%, 31.22 to 56.50 μg/mm^3^) were observed for the composite Filtek Supreme XT, and the lowest values for Grandio SO (0.28 to 0.72 wt%, 5.62 to 14.80 μg/mm^3^). An intermediate content was observed for Clearfil Majesty Posterior (0.54 to 0.90 wt%, 13.16 to 22.02 μg/mm^3^). All materials presented some water content before the water storage, which gradually increased during the water storage. Unlike the Filtek Supreme XT, no difference in the water content measured between 24 h and 48 h of storage was observed for Grandio SO and Clearfil Majesty Posterior. Except for the water content of Clearfil Majesty Posterior calculated in μg/mm^3^, no statistically significant water sorption was observed between 21 and 60 days of storage.

Figure 3 shows the water sorption, estimated according to material and storage time. The water sorption was calculated by subtracting the water content measured after each storage time from the values measured in non-stored specimens. Filtek Supreme XT absorbed the highest amount of water for all storage times, while the other composites evaluated showed similar water sorption values. After 60 days of storage, the highest water sorption values were observed for Filtek Supreme XT (25.3 μg/mm^3^), while the other composites absorbed approximately 9 μg/mm^3^. Even though the water sorption was similar, the final content observed for Grandio SO (0.7 wt%; 14.8 μg/mm^3^) was lower than that for Clearfil Majesty Posterior (0.9 wt%; 22.0 μg/mm^3^).

## 4. Discussion

In the present study, KFT quantified the number of water molecules released from the composite specimens, which constitute the water content. Therefore, the number of water molecules incorporated (water sorption) into specimens was estimated by subtracting the contents measured after the water storage from those of non-stored specimens. Interestingly, even though the non-stored Clearfil Majesty Posterior presented more than 2-fold water content than Grandio SO (5.62 vs. 13.16 μg/mm^3^), both composites presented similar water sorption during the storage in distilled water. The water content increased by approximately 3.0 μg/mm^3^ after 24 h of storage for both materials. Additional 8.9−9.2 μg/mm^3^ were added to that content observed in non-stored specimens when the specimens were maintained for 60 days in water. However, the water content increased (in μg/mm^3^) only for Clearfil Majesty Posterior between 21 and 60 days of storage, suggesting that this composite could be more prone to absorb water than Grandi SO. On the other hand, specimens of Filtek Supreme XT absorbed an average of 9.5 μg/mm^3^ after 24 h of water storage and reached 56.5 μg/mm^3^ (25.3 μg/mm^3^ more than non-stored specimens) after 60 days in water. Water sorption is strongly affected by the features of monomers, and the presence of more hydrophilic monomers (e.g., Bis-GMA) enhances the susceptibility of the polymer to absorb water. At the same time, a reduction is expected using more hydrophobic monomers such as TEGDMA [18]. All materials evaluated contain both hydrophilic and hydrophobic monomers, but the manufacturers do not fully disclose the percentage of each of these monomer blends. Therefore, explanations based on monomeric content would be merely speculative.

On the other hand, the filler content might explain some differences in water sorption among the materials evaluated. Increasing the filler content reduces the amount of polymer in composites and, consequently, the ability of the material to absorb water [13]. In the present study, Clearfil Majesty Posterior (82 vol%) has the highest filler content, and the Grandio SO is 11% lower filled (73 vol%) than the former material. Despite differences in ultimate water content, these composites showed similar water sorption values. Besides, the least filled composite Filtek Supreme XT (55.6 vol%) absorbed far more water than the other composites. Moreover, physical and chemical interactions between the matrix and the filler at the interphase strongly affected the water diffusion within the polymeric matrix [19]. Improper silanization of filler particles increases the hydrophilic surface area and, consequently, the pathway for water sorption [8]. Therefore, zirconia fillers containing composites tend to be more prone to absorb water because of the weak and unstable silane bond to zirconia [20,21]. The presence of zirconia fillers in the Filtek Supreme XT might also explain the highest water sorption observed for this material.

Water sorption values observed in the present study for Grandio SO are like those (14 μg/mm^3^ after 60 days in water) found using the ISO 4049 standard [14]. On the other hand, using KFT resulted in slightly lower values of water sorption than those measured with the ISO 4049 standard for the composites Clearfil Majesty Posterior (6.2 vs. 9.7 µg/mm^3^, reported by the manufacturer) and Filtek Supreme XT (21.4 vs. 29.1 µg/mm^3^) [22]. The water sorption is estimated based on specimens’ weight changes due to water storage using the ISO 4049 standard. Unlikely, KFT quantifies the number of water molecules incorporated into specimens. Therefore, it is reasonable to assume that this difference may be attributed to substances other than water that can also be incorporated into the specimen, modifying its weight.

The present study’s findings showed that all materials had water content ranging from 5.62 to 31.22 μg/mm^3^ (0.54 to 1.69 wt%) even before their storage in water. It would be reasonable to assume a certain error in base content assigned to water sorption upon specimen preparation under water lubrication, even for relatively short times, e.g., 15−30 min. However, a negative control measurement using the composite Grandio SO without any water contamination showed that its base content (4.3 ± 0.1 μg/mm^3^) was in the range of the water-lubricated processing followed by the described drying sequence (5.6 ± 0.1 μg/mm^3^). It can be concluded that the exposure of specimens to moist conditions for short periods might only increase the superficial free water content [23,24]. Therefore, this absorbed free water would be relatively volatile and released by keeping the specimens stored for one week in dry conditions. As a result, the base water content measured in our direct resin composite specimens is either incorporated during the manufacturing of composite pastes, or hygroscopically absorbed from surrounding humidity.

To the best of our knowledge, no prior study has evaluated the water content of direct dental composites using KFT. The Karl Fischer titration (KFT) is a reliable and feasible method to determine trace amounts of water in a specimen [16,21,24]. The titration is based on adding a titer (i.e., iodine) of a known concentration to a solution with unknown water content. Therefore, water contents are calculated by the amount of titer added until it reaches a balance in the reaction [25]. The coulometric KFT uses iodine generated by electrodes, and a constant alternating current is maintained until an excess of iodine is present in the titration cell [21]. This method accurately measures small water contents (<3%), but it is also essential to obtain an adequate water release from the specimens [25]. Several factors can affect the method’s accuracy, and pilot studies defined the parameters used in the present study. For instance, the oven temperature is a critical factor affecting the results. Temperatures lower than adequate can underestimate the water content values since bounded water is not accessed [16]. On the other hand, overheating the specimen may result in additional water production due to condensation reactions [16]. One of the pilot studies carried out by the Coulometer’s manufacturer gradually heated the dental composites from 50 to 250 °C. It was found that a temperature of 200 °C maximized the water release without any suggestive side reactions. Another critical matter is related to the specimens’ dimensions. The ISO 15512:2019 standard establishes that specimens with 0.2 to 0.4 g are recommended when the water content is expected to be between 0.5 and 1.0 wt% and weighing 0.1 to 0.2 g for samples containing more than 1.0 wt% of water [26]. The weight of the plate-shaped specimens with 100 mm^3^ was 0.184 ± 0.002 g for Filtek Supreme XT, 0.203 ± 0.002 g for Grandio SO, and 0.224 ± 0.002 g for Clearfil Majesty Posterior. Therefore, the specimens’ dimensions were in accordance with the ISO 15512:2019 standard, increasing the reliability of the results.

The KFT is an accurate and feasible method to measure the water content of composites. An important advantage of KFT over the ISO 4049 standard is that it allows determining the baseline and ultimate water content of specimens, and not only the water sorption of composites [11,12,13,14]. In the present study, despite similar water sorption following water storage, the ultimate water content of Clearfil Majesty Posterior was approximately 50% higher than that measured for Grandio SO. A method estimating the water sorption based on weight changes of specimens would be unable to show this final difference, and it could result in misconclusions [13]. In fact, water sorption of a resin composite in place of a clinical restoration is causing hydrolytic expansion. Negative consequences might be expected either towards mechanical polymer degradation or discoloration of a restoration [27]. Further effects of water sorption might be due to build-up of expansion stress in high C-factor cavities [28].

## 5. Conclusions

All direct resin composites evaluated presented some base water content, which increased following the water exposure. The highest base content was observed for Filtek Supreme XT composite and the lowest for Grandio SO. Except for Clearfil Majesty Posterior, no further water sorption was observed after 21 days of storage. Filtek Supreme XT composite absorbed more water than the other composites, which had similar water sorption values. However, the final water content of Clearfil Majesty Posterior was higher than that of Grandio SO.

## Figures and Tables

**Figure 1 materials-15-08524-f001:**
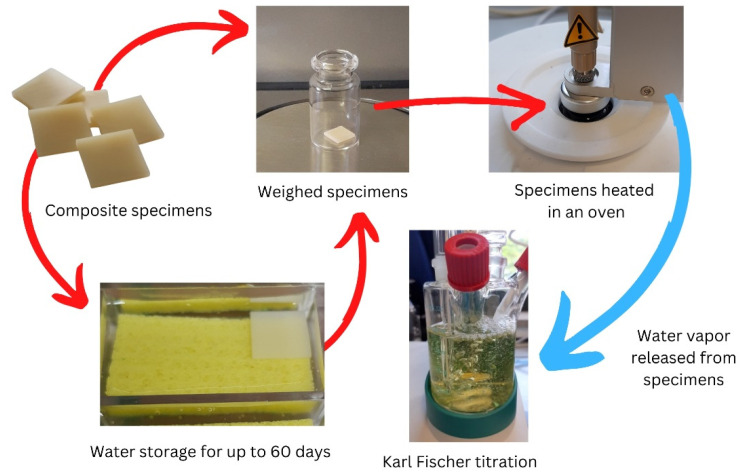
Diagram illustrating the experimental set-up.

**Figure 2 materials-15-08524-f002:**
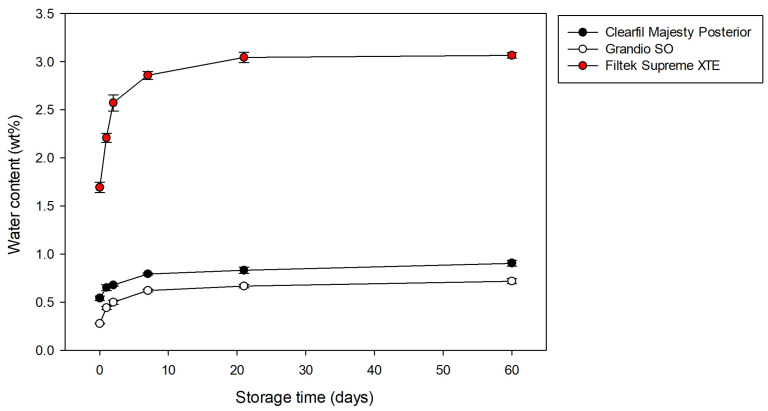
Means (standard deviation) of water content (in wt%) for each material evaluated as a function of storage times in water.

**Figure 3 materials-15-08524-f003:**
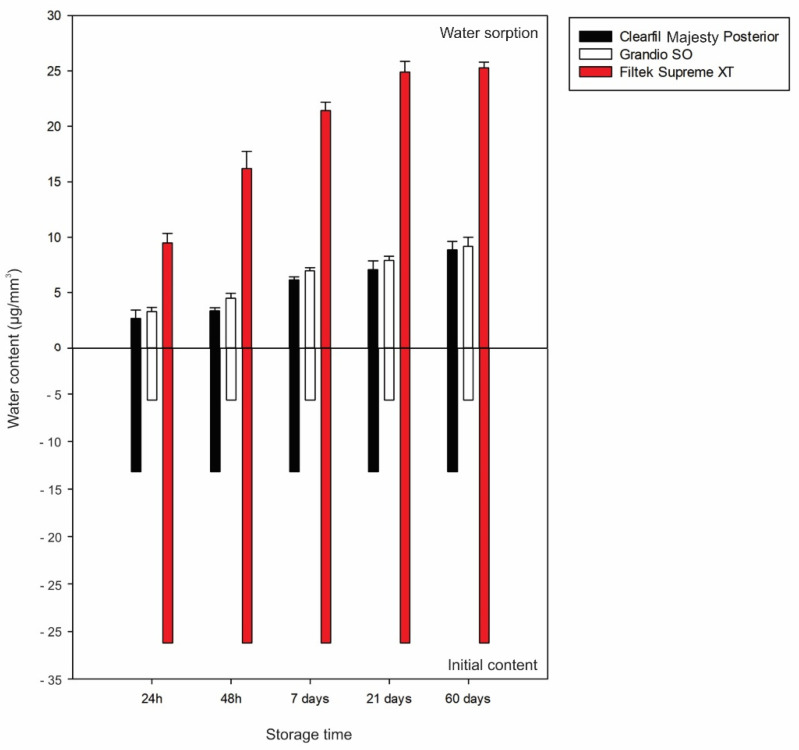
Bar charts illustrating the amount of water sorption (in μg/mm^3^) of each material according to storage times. The water sorption (upper part) was estimated by subtracting the final water content from the initial (lower part) measured for non-stored specimens.

**Table 1 materials-15-08524-t001:** Description of materials evaluated.

Material (Manufacturer)	Fillers	Vol% Filler	Matrix Monomers
Clearfil Majesty Posterior (Kuraray Noritake Dental Inc., Okyama, Japan)	Glass-ceramic, alumina micro, and silica fillers.	82	Bis-GMA, TEGDMA, and hydrophobic aromatic dimethacrylate.
Grandio SO (Voco, Cuxhaven, Germany)	Glass-ceramic fillers, and silicon dioxide nanoparticles.	73	Bis-GMA, Bis-EMA, and TEGDMA
Filtek Supreme XTE (3M ESPE, St. Paul, MN, USA)	Silica and zirconia fillers, and aggregated zirconia/silica cluster filler.	55.6	Bis-GMA, UDMA, TEGDMA, and Bis-EMA.

SiO_2_: silicium dioxide; ZrO_2_: zirconium dioxide; Bis-GMA: bisphenol A glycidyl methacrylate; Bis-EMA: ethoxylated bisphenol-A dimethacrylate; TEGDMA: triethylenglycol dimethacrylate; UDMA: urethane dimethacrylate.

**Table 2 materials-15-08524-t002:** Means (standard deviation) of water content measured in wt% according to material and storage time (*n* = 5).

Storage Time	Material		
	Clearfil Majesty Posterior	Grandio SO	Filtek Supreme XT
Non-stored	0.54 (0.02) ^Bd^	0.28 (0.00) ^Cd^	1.69 (0.05) ^Ae^
24 h	0.65 (0.03) ^Bc^	0.44 (0.02) ^Cc^	2.21 (0.05) ^Ad^
48 h	0.68 (0.01) ^Bc^	0.50 (0.02) ^Cc^	2.57 (0.08) ^Ac^
7 days	0.79 (0.01) ^Bb^	0.62 (0.01) ^Cb^	2.86 (0.04) ^Ab^
21 days	0.83 (0.03) ^Bab^	0.67 (0.02) ^Cab^	3.05 (0.05) ^Aa^
60 days	0.90 (0.03) ^Ba^	0.72 (0.03) ^Ca^	3.07 (0.03) ^Aa^

Means followed by the same letter (uppercase comparing materials, lowercase comparing storage times) have no statistical difference in Tukey’s test (*p* < 0.05).

**Table 3 materials-15-08524-t003:** Means (standard deviation) of water content measured in μg/mm^3^ according to material and storage time (*n* = 5).

Storage Time	Material		
	Clearfil Majesty Posterior	Grandio SO	Filtek Supreme XT
Non-stored	13.16 (0.56) ^Bd^	5.62 (0.08) ^Cd^	31.22 (0.99) ^Ae^
24 h	15.84 (0.75) ^Bc^	8.93 (0.35) ^Cc^	40.70 (0.86) ^Ad^
48 h	16.50 (0.30) ^Bc^	10.12 (0.44) ^Cc^	47.41 (1.55) ^Ac^
7 days	19.32 (0.27) ^Bb^	12.58 (0.28) ^Cb^	52.66 (0.76) ^Ab^
21 days	20.24 (0.78) ^Bb^	13.54 (0.36) ^Cab^	56.11 (0.98) ^Aa^
60 days	22.02 (0.74) ^Ba^	14.80 (0.82) ^Ca^	56.50 (0.52) ^Aa^

Means followed by the same letter (uppercase comparing materials, lowercase comparing storage times) have no statistical difference in Tukey’s test (*p* < 0.05).

## Data Availability

Data are available under request.

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
