# Peer review of "Determination of Water Content in Direct Resin Composites Using Coulometric Karl Fischer Titration"

_materials, 2022, doi:10.3390/ma15238524_

Round 1

Reviewer 1 Report

Nice idea and precise clear execution.  I would just suggest to add illustrative figure for the specimen preparation and testing method. In my opinion this may be helpful for more understanding.

Author Response

Response: Thanks for the time you`ve spent revising our manuscript. A diagram illustrating the experimental setup was added to the manuscript.

Reviewer 2 Report

The manuscript investigated the water content and sorption of direct composites over 60 days using coulometric Karl Fischer titration (KFT). The authors have performed some experiments as presented in this manuscript to investigate the water content and sorption. The study design was rigorous. The objectives of this study were clear, and the conclusion corresponded to the objectives. The paper can be improved by providing more convincing results to support the topic. Some concerns are listed as follows.

Major concerns

The author concluded that water content stabilized after 21 days. But from the results, no statistically significant water content was observed between 7 and 21 days of storage in Group CM and GS (Tables 1 and 2).

Other concerns

1.       In the abstract section, what does the “FT” mean?

2.       The author only provided some data in the result part of the abstract, but not in the main text. Please add the data.

3.       For the water content of Clearfil Majesty Posterior calculated in μg/mm3, statistically significant water sorption was observed between 21 and 60 days of storage (Table 3).  But for the water content of Clearfil Majesty Posterior calculated in wt%, no statistically significant water sorption was observed between 21 and 60 days of storage (Table 2).

Please justify.

4.       The weight of the plate-shaped specimens with 100 mm3 ranged from 0.184 ± 0.002 g for Filtek 225 Supreme XT to 0.224 ± 0.002 g for Clearfil Majesty Posterior. What was the weight of Grandio SO? Please justify

5.       It will be more straightforward to add a curve graph to show the trend of the results of the water content.

Author Response

The manuscript investigated the water content and sorption of direct composites over 60 days using coulometric Karl Fischer titration (KFT). The authors have performed some experiments as presented in this manuscript to investigate the water content and sorption. The study design was rigorous. The objectives of this study were clear, and the conclusion corresponded to the objectives. The paper can be improved by providing more convincing results to support the topic. Some concerns are listed as follows.

Response: Thank you very much for your consideration and the time you’ve spent reviewing our manuscript. We are grateful for helping us improve the manuscript.

The author concluded that water content stabilized after 21 days. But from the results, no statistically significant water content was observed between 7 and 21 days of storage in Group CM and GS (Tables 1 and 2).

Response: We stated that the stabilization occurred after 21 days because there are statistical differences between 7 (letter b, in both tables) and 60 days (letter a) for all composites. Otherwise, a statistical difference between 21 and 60 days was observed only for Clearfil Majesty Posterior. However, we speculate that this water content increase (less than 10%) is irrelevant.

In the abstract section, what does the “FT” mean?

Response: The mistype was revised.

“The water content of non-stored specimens ranged from 0.28 to 1.69 wt% (5.6 to 31.2 μg/mm3) for GS and FS, respectively.”

The author only provided some data in the result part of the abstract, but not in the main text. Please add the data.

Response: These data were added to the Results section accordingly.

“All materials presented some water content before the water storage, ranging from 0.28 to 1.69 wt% (5.6 to 31.2 μg/mm3) for Grandio SO and Filtek Supreme XT, respectively.”

“After 60 days of storage, the highest water sorption values were observed for Filtek Supreme XT (25.3 μg/mm3), while the other composites absorbed approximately 9 μg/mm3. Even though the water sorption was similar, the final content observed for Grandio SO (0.7 wt%; 14.8 μg/mm3) was lower than that for Clearfil Majesty Posterior (0.9 wt%; 22.0 μg/mm3).”

For the water content of Clearfil Majesty Posterior calculated in μg/mm3, statistically significant water sorption was observed between 21 and 60 days of storage (Table 3).  But for the water content of Clearfil Majesty Posterior calculated in wt%, no statistically significant water sorption was observed between 21 and 60 days of storage (Table 2). Please justify.

Response: The coulometer provided the results in wt%, which was converted to μg/mm3 based on weight and specimen dimensions. We hand-measured the dimensions using a digital caliper and expected minor errors in μg/mm3 values could explain the slight differences observed in statistical results.

The weight of the plate-shaped specimens with 100 mm3 ranged from 0.184 ± 0.002 g for Filtek 225 Supreme XT to 0.224 ± 0.002 g for Clearfil Majesty Posterior. What was the weight of Grandio SO? Please justify

Response: The weight of Grandio SO specimens was 0.203 ± 0.002 g (added to the manuscript).

“The weight of the plate-shaped specimens with 100 mm3 was 0.184 ± 0.002 g for Filtek Supreme XT, 0.203 ± 0.002 g for Grandio SO, and 0.224 ± 0.002 g for Clearfil Majesty Posterior.”

The composition differences justify discrepancies in densities among the materials. But, since the exact composition of commercially available materials is unknown, any discussion would be merely speculative.

It will be more straightforward to add a curve graph to show the trend of the results of the water content.

Response: A graphic (Figure 2) showing the water content tendency was added to the Results section.

Reviewer 3 Report

This investigation is a paper that presents information for clinicians and researchers in the field of restorations with resin composite used in dentistry due to their adequate esthetical and mechanical properties to restore anterior and posterior teeth.

Abstract. This section is correct and shows the main aspects of the paper.

Introduction. Several studies have evaluated the ability of dental composites to absorb water than can compromise the mechanical performance of the composite and favor the occurrence of fractures in the restoration. The aim of this study was the present study aimed to measure the initial water content and amount of water absorbed by direct resin composites using a coulometric method.

In the third paragraph between 46-53 lines, the text must be changed to the section Discussion because reported information about the used method

Material and methods. The authors must report if the research protocol is an original research. In this section. Each step of the protocol must report the related references. This aspect is very important.

Results. This section is correct and shows the water sorption estimated according to material and storage time. The water sorption was calculated by subtracting the water content measured after each storage time by the values measured in non-stored specimens.

Discussion. The discussion is correct, but the last paragraph must report some references.

Conclusions. This is too short. It must include some main aspects of the paper.  

References. There are too old references. 41.3% references are papers with more of 5 years.

There are three related references with proceedings of congress (19,23,28). The authors must report references of Pubmed  

Conclusively, the study is not ready for publication and must be revised.

Author Response

This investigation is a paper that presents information for clinicians and researchers in the field of restorations with resin composite used in dentistry due to their adequate esthetical and mechanical properties to restore anterior and posterior teeth.

Abstract. This section is correct and shows the main aspects of the paper.

Introduction. Several studies have evaluated the ability of dental composites to absorb water than can compromise the mechanical performance of the composite and favor the occurrence of fractures in the restoration. The aim of this study was the present study aimed to measure the initial water content and amount of water absorbed by direct resin composites using a coulometric method.

In the third paragraph between 46-53 lines, the text must be changed to the section Discussion because reported information about the used method.

Response: This part of the text was placed in the Discussion section, as suggested by the reviewer.

Material and methods. The authors must report if the research protocol is an original research. In this section. Each step of the protocol must report the related references. This aspect is very important.

Results. This section is correct and shows the water sorption estimated according to material and storage time. The water sorption was calculated by subtracting the water content measured after each storage time by the values measured in non-stored specimens.

Discussion. The discussion is correct, but the last paragraph must report some references.

Response: We added some references to the last paragraph of the Discussion section.

Fugolin A.P.; de Paula, A.B.; Dobson, A.; Huynh, V.; Consani, R.; Ferracane, J.L.; Pfeifer, C.S. Alternativemonomer for BisGMA-free resin composites formulations. Dent Mater 2020, 36, 884-892. https://doi.org/10.1016/j.dental.2020.04.009

Fugolin, A.P.; Lewis. S.; Logan. M.G.; Ferracane, J.L.; Pfeifer, C.S. Methacrylamide-methacrylate hybrid monomers for dental applications. Dent Mater 2020, 36, 1028-1037. https://doi.org/10.1016/j.dental.2020.04.023

Wendler, M.; Stenger, A.; Ripper, J.; Priewich, E.; Belli. R.; Lohbauer, U. Mechanical degradation of contemporary CAD/CAM resin composite materials after water ageing. Dent Mater 2021, 37, 1156-1167. https://doi.org/10.1016/j.dental.2021.04.002

ISO 4049. Dentistry-polymer-based filling, restorative and luting materials. 2019

Gauthier, R.; Aboueillei, H.; Boussès,Y.; Brulat-Bouchard, N.; Colon, P.; Chenal, J.M.; Tillier, Y.; Grosgogeat, B. Experimental investigation of dental composites degradation after early water exposure. Int. J Thermophys 2022, 9, 1-18. https://doi.org/10.1115/1.4056197

Suiter, E.A.; Watson, L.E.; Tantbirojn, D.; Lou, J.S.; Versluis, A. Effective expansion: balance between shrinkage and hygroscopic expansion. J Dent Res 2016, 95, 543-549. https://doi.org/10.1177/0022034516633450 

Conclusions. This is too short. It must include some main aspects of the paper.  

Response: The conclusion was improved accordingly.

“All direct resin composites evaluated presented some base water content, which increased following the water exposure. The highest base content was observed for Filtek Supreme XT composite and the lowest for Grandio SO. In general, the water content stabilized after 21 days of storage. Filtek Supreme XT composite absorbed more water than the other composites, which had similar water sorption values. However, the final water content of Clearfil Majesty Posterior was higher than that of Grandio SO.”

References. There are too old references. 41.3% references are papers with more of 5 years. There are three related references with proceedings of congress (19,23,28). The authors must report references of Pubmed.

Response: The references were updated accordingly. However, some references added were indexed to the Scopus database but not to Pubmed. The new references are:

Aro, R.; Ben Ayoub, M.W.; Leito, I.; Georgin, E. Moisture in solids: comparison between evolved water vapor and vaporization coulometric Karl Fischer methods. Int J Thermophys 2020, 41: 113. https://doi.org/10.1007/s10765-020-02697-6

Hashemian, A.; Shahabi, S.; Behroozibakhsh, M.; Najafi, F.; Abdulrazzaq Jerri Al-Bakhakh, B.; Hajizamani, H. A modified TEGDMA-based resin infiltrant using polyurethane acrylate oligomer and remineralising nano-fillers with improved physical properties and remineralisation potential. J Dent 2021, 113, 103810. https://doi.org/10.1016/j.jdent.2021.103810  

Martim, G.C.; Kupfer, V.L.; Moisés, M.P.; dos Santos, A.; Buzzetti, P.H.M.; Rinaldi, A.W.; Rubira, A.F.; Girotto, E.M. Physical-chemical properties of dental composites and adhesives containing silane-modified SBA-15. J Mech Behav Biomed Mater 2018, 80, 277-284. https://doi.org/10.1016/j.jmbbm.2018.02.009

Chalykh, A.E.; Petrova T.F.; Ponomarev, I.I. Water sorption by polyheteroarylenes. Polymers 2022, 14, 2255. https://doi.org/10.1016/10.3390/polym14112255

Aro, R., Ayoub, M.W.B., Leito, I., Georgin, É., Savanier, B. Calibration and uncertainty estimation for water content measurement in solids. Int J Thermophys 2021, 42, 42. https://doi.org/10.1007/s10765-021-02796-y

Gauthier, R.; Aboueillei, H.; Boussès ,Y.; Brulat-Bouchard, N.; Colon, P.; Chenal, J.M.; Tillier, Y.; Grosgogeat, B. Experimental investigation of dental composites degradation after early water exposure. J Biomech Eng 2022, 9, 1-18. https://doi.org/10.1115/1.4056197

Suiter, E.A.; Watson, L.E.; Tantbirojn, D.; Lou, J.S.; Versluis, A. Effective expansion: balance between shrinkage and hygroscopic expansion. J Dent Res 2016, 95, 543-549. https://doi.org/10.1177/0022034516633450 

Conclusively, the study is not ready for publication and must be revised.

Response: Thank you very much for your consideration and your time reviewing our manuscript. We hope the revised manuscript fulfills the criteria for publication in this prestigious journal.
